# Reconstruction of Single-Cell Trajectories Using Stochastic Tree Search

**DOI:** 10.3390/genes14020318

**Published:** 2023-01-26

**Authors:** Jingyi Zhai, Hongkai Ji, Hui Jiang

**Affiliations:** 1Department of Biostatistics, University of Michigan, Ann Arbor, MI 48109, USA; 2Department of Biostatistics, Johns Hopkins Bloomberg School of Public Health, Baltimore, MD 21205, USA

**Keywords:** trajectory reconstruction, single-cell RNA sequencing data analysis, embedding location tree

## Abstract

The recent advancement in single-cell RNA sequencing technologies enables the understanding of dynamic cellular processes at the single-cell level. Using trajectory inference methods, pseudotimes can be estimated based on reconstructed single-cell trajectories which can be further used to gain biological knowledge. Existing methods for modeling cell trajectories, such as minimal spanning tree or k-nearest neighbor graph, often lead to locally optimal solutions. In this paper, we propose a penalized likelihood-based framework and introduce a stochastic tree search (STS) algorithm aiming at the global solution in a large and non-convex tree space. Both simulated and real data experiments show that our approach is more accurate and robust than other existing methods in terms of cell ordering and pseudotime estimation.

## 1. Introduction

The advancement of single-cell RNA sequencing enables measuring of gene expression for individual cells to prompt an understanding of dynamic cellular processes, including cell state transitions such as cell differentiation. Reconstructing a cell trajectory from the gene expression for a sample of cells is one new research area made possible by this technology. However, the high-dimensional gene expression data space and the associated high-level noise pose difficulties in modeling the trajectory from the original expression data [1]. One way to reconstruct the cell trajectory is by the calculation of pseudotime, where pseudotime is a measure of the distance of a particular cell from the origin in a dynamic process. This type of computational approach is called trajectory reconstruction (TR) [2]. To overcome the challenges in single-cell trajectory analysis, TR methods generally have two main steps: First, to handle the high dimensionality and high noise level in the expression data, a dimensionality reduction method is applied to convert the original high-dimensional data space into a low-dimensional space. Both linear and nonlinear dimensionality methods can be considered in this step to address different types of data. The second step is to model the trajectory in the dimension-reduced space. For instance, a minimal spanning tree (MST) is fitted or the k-nearest neighbor (KNN) graph is applied to model the cell trajectory [3]. This paper focuses on the second trajectory-modeling approach, to further improve the performance of the TR method in the reconstruction of single-cell trajectories.

The existing TR methods can be classified into three main categories based on the trajectory-modeling steps. First, Wanderlust, Wishbone and SLICER are all designed based on the KNN graph. Wanderlust was not originally developed for single-cell transcriptomics data but for cytometry data, so a dimensionality reduction is implemented and the trajectory-modeling is applied directly to the high-dimensional data space [4]. Based on Wanderlust, Wishbone adds a dimensionality reduction step before trajectory modeling to address the high-dimensionality challenge in the scRNA-seq data. Unlike Wanderlust, Wishbone can also detect a bifurcating event with two branches [5]. Similarly to Wishbone, SLICER fits the KNN graph on a lower dimension but with a different dimension reduction approach—LLE. SLICER requires less prior information than the previous two approaches and is also able to detect “bubbles”—a special type of cell trajectory [6]. Secondly, several TR methods, such as Monocle, TSCAN and Slingshot, use an MST to model a cell trajectory. Monocle is a pioneering method which uses ICA to reduce dimensionality [7]. TSCAN further reduces the complexity in the data structure by a model-based clustering algorithm, where an MST is obtained by connecting the cluster centers [8]. More recently, Slingshot also fits an MST on clusters of cells similar to TSCAN, but the MST is treated as the initial guess for the simultaneous principal curve algorithm [9]. Third, in addition to the KNN graph and an MST, other trajectory-modeling methods can also be employed. For instance, diffusion pseudotime (DPT) uses diffusion-like random walks to model transitions between cells. DPT is more computationally efficient and suffers less from overfitting issues caused by the dimensional reduction step. However, DPT only models a bifurcating trajectory and does not provide an explicit tree structure [10]. As an advanced version of Monocle, Monocle2 first reduces dimensionality by the “dpFeature” method. Then, a group of centroids are obtained as the latent representation of the dimension-reduced data by the soft k-means clustering, and a spanning tree is fitted to the latent data. Finally, Monocle2 uses reversed graph embedding (RGE) to obtain a principal graph [11].

However, the majority of these TR methods use gradient approaches in the trajectory-modeling step; thus, non-convex optimization is involved with a locally searching algorithm applied [2]. With a locally searching algorithm, there is a risk that the algorithm becomes stuck in the local solution and outputs it as the final solution. Unlike these TR methods, our proposed stochastic tree search (STS) algorithm searches for the optimal solution globally. We first construct a penalized likelihood and then apply a stochastic optimization algorithm to search through the tree space and obtain the final optimal solution. Thus, we can identify an optimal tree directly in the lower dimension converted by a dimensionality reduction method. Moreover, as the objective function in the optimization is flexible, STS is applicable to more types of data than those MST-based approaches.

The remaining parts of the paper are organized as follows: In Section 2, we introduce our trajectory-modeling method with the dynamic optimization in the tree searching space. Section 3 presents a simulation study to compare our algorithm with four other methods in different settings. In Section 4, the implementation of our method is shown through two single-cell RNA sequencing datasets as well as in a comparison with four existing methods. Section 5 concludes the paper with a discussion.

## 2. Methods

Similar to most existing TR methods, our single-cell trajectory reconstruction approach also consists of two parts. The first part is flexible with any dimensionality reduction method, while we develop a novel stochastic tree searching process to estimate the cell trajectory for the second part. Under the likelihood framework, we define an optimization function used to find the optimal embedding tree as the estimated cell trajectory. During the stochastic tree searching process, we search through the whole embedding tree space with a pool of candidate trees at each time and start from the simplest one-edge tree structure to a more complicated tree structure with more edges. The details of our stochastic tree searching algorithm are shown in the following subsections.

### 2.1. Preprocessing

Before fitting the tree structure on the data, the raw gene read counts are normalized by log2 transformation. Then, the dimension of normalized data is reduced by any linear or nonlinear method. Principal component analysis (PCA) applies a linear projection of the data, which preserves the variance in the new lower dimension space. Locally linear embedding (LLE), diffusion maps, and t-SNE are more general approaches without the linear relationship assumption, so these methods are able to find nonlinear relationships between cells. The data drive the choice of dimensionality reduction approach. If the linear assumption holds in the data, PCA will be applied since it is relatively computational efficient [1]. Otherwise, nonlinear methods will be applied, especially for the more recent data set with a more complex cell trajectory.

### 2.2. Penalized Likelihood

We use an embedding tree on the lower-dimension space to estimate the cell trajectory. An embedding tree T=(V,E,y) can be defined with three main components V, E and y. V is a vertex set of size |V|, and E is an edge set with E⊂V×V. $y=(y_{1},...,y_{\left|V\right|})$ is the associated vertex embedding, where $y_{j}\in R^{p}$. When we fit the embedding tree T with n data points $X=\{X_{1},...,X_{n}\}$, we assume that
$$X_{i}|T∼N\left(\phi_{T}\left(X_{i}\right),\;\sigma^{2}I\right),$$
where $X_{i}\in R^{p}$ and $\phi_{T}\left(X_{i}\right)$ is the projection of $X_{i}$ to the embedding tree T. Now we have
$$p\left(X_{i}|T\right)={\left(2\pi \sigma^{2}\right)}^{-\frac{p}{2}}\exp\left\{-\frac{1}{2\sigma^{2}}{\left(X_{i}-\phi_{T}\left(X_{i}\right)\right)}^{T}\left(X_{i}-\phi_{T}\left(X_{i}\right)\right)\right\}.$$
which leads to the likelihood function
(1)$$\begin{array}{cc} L(T;X) & ={\left(2\pi \sigma^{2}\right)}^{-\frac{np}{2}}\exp\{-\frac{1}{2\sigma^{2}}\sum_{i=1}^{n}\left|\right|X_{i}-\phi_{T}\left(X_{i}\right){\left|\right|}^{2}\}\\ \; & \propto \exp\{-\frac{1}{2\sigma^{2}}\sum_{i=1}^{n}||X_{i}-\phi_{T}\left(X_{i}\right)|{|}^{2}\} \end{array}$$
where we assume that $\sigma^{2}$ is known and can be estimated from the data. The negative log-likelihood function is
(2)$$\begin{array}{cc} -l & =-\log\left(L\right)=\frac{1}{2\sigma^{2}}\sum_{i=1}^{n}\left|\right|X_{i}-\phi_{T}\left(X_{i}\right){\left|\right|}^{2}\\ \; & \propto \sum_{i=1}^{n}\left|\right|X_{i}-\phi_{T}\left(X_{i}\right){\left|\right|}^{2} \end{array}$$

Finally, we can find the optimal embedding tree by minimizing the following penalized negative log-likelihood function,
$$\min_{T}f\left(T\right)+p\left(T\right).$$f(T) is the loss function derived from the negative log-likelihood function as follows,
$$f\left(T\right)=\sum_{i=1}^{n}\left|\right|x_{i}-\phi_{T}\left(x_{i}\right){\left|\right|}^{2},$$
where $\phi_{T}\left(x_{i}\right)$ is the projection of $x_{i}$ to the embedding tree T. The penalty term p(T) has two components as follows,
$$p\left(T\right)=\lambda_{1}{\left|V\right|}^{\alpha }+\lambda_{2}\sum_{e=(j,k)\in E}\left|\right|y_{j}-y_{k}{\left|\right|}^{2},\;\lambda_{1},\lambda_{2},\alpha >0.$$The first part in the penalty term controls the complexity of the tree structure. With the larger tree size, more penalties will be added to the optimization function. Based on the BIC
$$BIC=K\log\left(n\right)-2\log\left(\hat{L}\right)\approx \frac{1}{{\hat{\sigma }}^{2}}\sum_{i=1}^{n}\left|\right|x_{i}-\phi_{T}\left(x_{i}\right){\left|\right|}^{2}+p\left|V\right|\log\left(n\right)$$
where ${\hat{\sigma }}^{2}$ is the sample variance. *K* is the number of free parameters for the tree model, so we have K≈p|V|. Therefore, we set $\lambda_{1}$ and α as follows,
$$\lambda_{1}={\hat{\sigma }}^{2}\log\left(n\right)p,\;\alpha =1,$$The second part shrinks the length of tree edges. We use a small penalty term with $\lambda_{2}=0.01$ to avoid any unnecessary long edges.

### 2.3. Projection of the Data to the Tree

In order to obtain the penalized negative log-likelihood function for the data, we first introduce the way to calculate the projection of each data point to a given tree. For an embedding tree T=(V,E,y), the projection of a data point $x_{i}$ to the tree T is defined as $\phi_{T}\left(x_{i}\right)$—a point on the tree T such that $\left|\right|x_{i}-\phi_{T}\left(x_{i}\right)\left|\right|$ is minimized. We further denote $\phi_{e}\left(x_{i}\right)$ as the project of $x_{i}$ to an edge $e=(V_{j},V_{k})\in E$. Here, *e* is an edge connecting two vertices $V_{j},V_{k}\in V$ with the associated embedding $y_{j},y_{k}\in R^{p}$, where j,k∈{1,...,|V|}. We can further define $\phi_{e}\left(x_{i}\right)$ in the following way,
$$\phi_{e}\left(x_{i}\right)=y_{j}+l_{e}\left(x_{i}\right)\left(y_{k}-y_{j}\right),$$
and
$$l_{e}\left(x_{i}\right)=\underset{l\in [0,1]}{\mathrm{argmin}}\left|\right|x_{i}-\left[y_{j}+l\left(y_{k}-y_{j}\right)\right]\left|\right|,$$
where $l_{e}\left(x_{i}\right)$ is the mapping of $\phi_{e}\left(x_{i}\right)$ when mapping the line segment $\left[y_{j},y_{k}\right]$ to [0,1] which can be calculated using the following closed-form formula
$$l_{e}\left(x_{i}\right)=\max\left\{0,\min\left(1,\frac{<x_{i}-y_{j},y_{k}-y_{j}>}{\left|\right|y_{k}-y_{j}{\left|\right|}^{2}}\right)\right\}.$$We have
$$\phi_{T}\left(x_{i}\right)=\min_{\phi_{e}\left(x_{i}\right)}\left|\right|x_{i}-\phi_{e}\left(x_{i}\right)\left|\right|$$
and
$$\left|\right|x_{i}-\phi_{T}\left(x_{i}\right)\left|\right|=\min_{e}\left|\right|x_{i}-\phi_{e}\left(x_{i}\right)\left|\right|.$$

### 2.4. Updating Vertex Embedding Location

In the optimization process, we start with a random tree and then iteratively project the data to the tree (detailed in the section above) and update the embedding locations of its vertices. Without loss of generality, suppose there is a vertex v∈V and $\left\{v_{1},v_{2},...,v_{m}\right\}$ is the set of all the vertices that *v* connects to. Let $x_{k_{1}},...,x_{k_{n_{k}}}\in R^{p}$ be the data points projected to edge $e_{k}=\left(v_{k},v\right)$ for k=1,...,m. The new embedding location $y^{*}$ for *v* can be calculated through minimizing the objective function
$$S\left(y\right)=\sum_{i=1}^{n}\left|\right|x_{i}-\phi_{T}\left(x_{i}\right){\left|\right|}^{2}+\lambda_{2}\sum_{k=1}^{m}||y-y_{k}{\left|\right|}^{2}.$$Let $l_{k_{t}}=l_{e_{k}}\left(x_{k_{t}}\right)$ be the mapping of $\phi_{e_{k}}\left(x_{k_{t}}\right)$ when mapping $\left[y_{k},y\right]$ to [0,1], and $y^{*}$ can then be computed by the following formula,
$$y^{*}=\underset{y}{\mathrm{argmin}}\sum_{k}\sum_{t}\left|\right|x_{k_{t}}-\left[y_{k}+l_{k_{t}}\left(y-y_{k}\right)\right]{\left|\right|}^{2}+\lambda_{2}\sum_{k}||y-y_{k}{\left|\right|}^{2}.$$If we assume that the mapping $l_{k_{t}}$ is fixed for $y^{*}$, there is a closed form solution for updating $y^{*}$. See Appendix A for more details. However, the mapping $l_{k_{t}}$ will change as the embedding location $y^{*}$ is updated. Hence, there is no simple closed-form solution for $y^{*}$ and we apply a backtracking line search algorithm to obtain $y^{*}$ as follows. We first set the initial values $y^{0}$ and $\tau^{0}=\tau_{max}=1$. Then, for the *b*-th iteration, b=1,…,B, we take the following steps:1.$\tau^{b}=p\tau^{b-1}$2.$y^{b+1}=y^{b}-\tau^{b}\nabla S_{b}$3.Continue the iteration until we have $S\left(y^{b}-\tau^{b}\nabla S_{b}\right)-S\left(y^{b}\right)\leq c\tau^{b}\nabla S_{b}^{T}\nabla S_{b}$.

Here there is no simple closed-form solution for the gradient $\nabla S_{b}$ as the mapping $l_{k_{t}}$ is also related to $y^{*}$. Therefore, we calculate the gradient $\nabla S_{b}$ by the numerical method where $\nabla S_{b}\approx \frac{S\left(y^{b}+\epsilon I\right)-s(\left(y^{b}\right)}{\epsilon }$. We use a small ϵ = 1 × 10${}^{-6}$ in the previous numerical approximation formula. We use p=0.8 and c=0.5 in our simulation studies. Finally we repeat the above backtracking line search process until the difference in the objective function reaches the tolerance.

### 2.5. Tree Similarity Score

To facilitate the stochastic tree search, we maintain a pool of candidate trees. During the stochastic tree searching process, we remove similar trees and therefore maintain the diversity of all candidate trees in the tree pool, which is achieved using a tree similarity score. The tree similarity score is defined to estimate the similarity between two trees of the same size. Considering two embedding trees $T_{1}=\left(V^{\left(1\right)},E^{\left(1\right)},y^{\left(1\right)}\right)$ and $T_{2}=\left(V^{\left(2\right)},E^{\left(2\right)},y^{\left(2\right)}\right)$ with the same size $|V^{\left(1\right)}\left|=\right|V^{\left(2\right)}|=s$, the tree similarity score can be computed as follows,
$$S\left(T_{1},T_{2}\right)=D\left(V^{\left(1\right)}\left(y^{\left(1\right)}\right),V^{\left(2\right)}\left(y^{\left(2\right)}\right)\right)+D\left(E^{\left(1\right)},E^{\left(2\right)}\right).$$The first part $D(V^{\left(1\right)}\left(y^{\left(1\right)}\right),V^{\left(2\right)}\left(y^{\left(2\right)}\right))$ denote the similarity between the two vertex sets $V^{\left(1\right)}$ and $V^{\left(2\right)}$ through the embedding location $y^{\left(1\right)}$ and $y^{\left(2\right)}$ as follows,
$$D\left(V^{\left(1\right)}\left(y^{\left(1\right)}\right),V^{\left(2\right)}\left(y^{\left(2\right)}\right)\right)=\sum_{i}^{s}\min_{j}\left|\right|y_{i}^{\left(1\right)}-y_{j}^{\left(2\right)}\left|\right|.$$The second part $D(E^{\left(1\right)},E^{\left(2\right)})$ calculates the distance between two edge sets $E^{\left(1\right)}$ and $E^{\left(2\right)}$ as
$$D\left(E^{\left(1\right)},E^{\left(2\right)}\right)=\sum_{i}^{s-1}\sum_{j}^{s-1}D\left(E_{i}^{\left(1\right)},E_{j}^{\left(2\right)}\right)=\sum_{i}^{s-1}\sum_{j}^{s-1}\sqrt{{\left\{D_{ij}^{\left(1\right)}\right\}}^{2}+{\left\{D_{ij}^{\left(2\right)}\right\}}^{2}},$$
where $E_{i}^{\left(1\right)}=\left(V_{i1}^{\left(1\right)},V_{i2}^{\left(1\right)}\right)$, $E_{j}^{\left(2\right)}=\left(V_{j1}^{\left(2\right)},V_{j2}^{\left(2\right)}\right)$ and
$$\left|\right|E_{i}^{\left(1\right)}-E_{j}^{\left(2\right)}\left|\right|=\left[\begin{array}{c} D_{ij}^{\left(1\right)}\\ D_{ij}^{\left(2\right)} \end{array}\right]=\left[\begin{array}{c} \min\{||y_{i1}^{\left(1\right)}-y_{j1}^{\left(2\right)}||,||y_{i1}^{\left(1\right)}-y_{j2}^{\left(2\right)}||\}\\ \min\{||y_{i2}^{\left(1\right)}-y_{j1}^{\left(2\right)}||,||y_{i2}^{\left(1\right)}-y_{j2}^{\left(2\right)}||\} \end{array}\right].$$

### 2.6. Stochastic Optimization

#### 2.6.1. Initial Tree Generation

Starting from the most simple tree structure—one-edge tree, the associated embedding locations are generated from a multivariate normal distribution MVN(μ,Σ). The mean vector μ and diagonal variance matrix Σ can be estimated from the data. To better search through the large tree space, multiple initial trees $\left\{T_{2}^{1},...,T_{2}^{L_{1}}\right\}$ are generated to form the initial tree pool $\left\{T_{2}^{l}\right\}$. We then update the embedding location for each tree in the initial tree pool to better fit the data. For each updated tree $T_{2}^{*}$, we calculate the tree similarity score $S_{1}$ and optimization score $S_{2}$ as follows,
$$S_{1}\left(T_{2}^{*}\right)=\min_{T_{2}\in \left\{T_{2}^{l}\right\}}S\left(T_{2},T_{2}^{*}\right),$$
$$S_{2}\left(T_{2}^{*}\right)=-\sum_{i=1}^{n}\left|\right|x_{i}-\phi_{T_{2}^{*}}\left(x_{i}\right){\left|\right|}^{2}+\lambda_{1}\left|V\right|+\lambda_{2}\sum_{e=(j,k)\in E}\left|\right|y_{j}-y_{k}{\left|\right|}^{2}.$$Let $T_{2}^{min}$ denote the tree in the pool where $S\left(T_{2}^{min},T_{2}^{*}\right)=\min_{T_{2}\in \left\{T_{2}^{l}\right\}}S\left(T_{2},T_{2}^{*}\right)$, and *a* is a pre-defined cutoff value. If $S_{1}\left(T_{2}^{*}\right)>a$ and $S_{2}\left(T_{2}^{*}\right)<\min_{T_{2}\in \left\{T_{2}^{l}\right\}}S_{2}\left(T_{2}\right)$, then we update the current initial tree by replacing $T_{2}^{min}$ with $T_{2}^{*}$.

#### 2.6.2. Grow Trees by Adding Nodes

Based on the old tree pool $\left\{T_{k}^{l}\right\}$ where we have all candidate trees with the tree size equal to *k*, we grow the trees by adding a new node to each tree. There are two ways to add a new node. The first one is adding a new node $v_{new}$ connected to any existing node $v_{j},j\in 1,\cdots ,k$ of $T_{k}$ and obtaining the new edge $e_{new}=\left(v_{new},v_{j}\right)$. The probability of selecting any node among all nodes of $T_{k}$ is set as follows,
$$P\left(e_{new}=\left(v_{new},v_{j}\right)\right)=\frac{1}{k}.$$Here, we assume there is an equal probability of selecting each existing node to connect with the new node. In order to make the stochastic optimization more efficient, we generate the new embedding location $y_{new}$ of $v_{new}$ based on the guidance from the data. In detail, we first compute the residuals based on $T_{k}$,
$$r_{i}=\left|\right|x_{i}-\phi_{T_{k}}\left(x_{i}\right){\left|\right|}^{2}.$$Then, the residuals are standardized as
$$r_{i}^{*}=\frac{r_{i}}{max_{i}\left\{r_{i}\right\}}.$$Finally, we sample the embedding location of the new node $y_{new}$ from $x_{1},...,x_{n}$ with the sampling weights $w_{i}=e^{\lambda_{3}r_{i}^{*}}$. The new node is more likely to locate in the area with larger residuals since they have a higher sampling probability. The parameter $\lambda_{3}$ controls the level of new embedding location driven by residuals.

The other way to generate a new node is by adding a middle point to any edge $e=(v_{j},v_{g})$ of $T_{k}$. Each edge of $T_{k}$ has the same probability to be chosen to add a middle point, where
$$P\left(e=\left(v_{j},v_{g}\right)\right)=\frac{1}{k-1},j\neq g\in \left\{1,...,k\right\}.$$We replace the previous edge *e* with two new edges $e_{1}^{new}=\left(v_{j},v_{new}\right)$ and $e_{2}^{new}=\left(v_{g},v_{new}\right)$. The new embedding location $y_{new}$ can be calculated by the following formula,
$$y_{new}=\frac{1}{2}\left(y_{j}+y_{g}\right),$$
where $y_{j}$ and $y_{g}$ are the embedding locations for $v_{j}$ and $v_{g}$.

#### 2.6.3. Optimizing Tree with Data

From the old tree pool $\left\{T_{k}^{l}\right\}$, we can obtain a new tree pool $\left\{T_{k+1}^{l}\right\}$ by adding a new node as described in the previous section. For each new tree, we further update the embedding location for all nodes to better fit the data. Similar to the initial tree updating procedure, for each updated tree $T_{k+1}^{*}$ in the optimization process, we compute the tree similarity score $S_{1}$ and optimization score $S_{2}$. If $S_{1}\left(T_{k+1}^{*}\right)>a$ and $S_{2}\left(T_{k+1}^{*}\right)<\min_{T_{k+1}\in \left\{T_{k+1}^{l}\right\}}S_{2}\left(T_{k+1}\right)$, then we update the current initial tree by replacing $T_{k+1}^{min}$ with $T_{k+1}^{*}$, where $T_{k+1}^{min}$ is the tree in the pool with the smallest tree similarity score $S(T_{k+1}^{*},T_{k+1})$. We repeat adding a new node step on the sequence of old trees for $L_{2}$ times and update the existing new tree pool $\left\{T_{k+1}^{l}\right\}$ according to the above two criteria.

#### 2.6.4. Final Optimal Tree

According to the optimization score $S_{2}\left(T_{k}\right)$, we order trees in each tree pool $\left\{T_{k}^{l}\right\}$, for k=2,..., and then we select the top ranked tree with smallest optimization score $S_{2}\left(T_{k}^{opt}\right)=\min_{T_{k}\in \left\{T_{k}^{l}\right\}}S_{2}\left(T_{k}\right)$. When $S_{2}\left(T_{k+1}^{opt}\right)>S_{2}\left(T_{k}^{opt}\right)$, we stop developing trees and obtain the final optimal tree $T^{opt}=T_{k}^{opt}$. The details of the process of finding the final optimal tree are shown in Figure 1.

### 2.7. Pseudotime Calculation

Once the optimal tree is found through the above stochastic tree searching method, we can then compute pseudotime through the same shortest path algorithm as applied by Bendall (2014) in the pseudotime calculation for Wanderlust [12]. We first define the length of an edge as the Euclidean distance between the two vertices on this edge, so that a corresponding adjacency matrix for the optimal tree can be calculated. Then, the distance between each cell to the origin is computed by the shortest path algorithm. Finally, we define the distance as the pseudotime for cells and order cells by their pseudotimes.

### 2.8. Extension from Linear Trees to Nonlinear Trees

The assumption of a linear tree-shaped cell trajectory is not always valid for most scRNA-seq data sets. The allowance of nonlinear cell trajectory estimation can improve the accuracy in the reconstruction of a cell trajectory. Hence, we also propose an improved the global searching algorithm—a stochastic tree searching algorithm to include nonlinear cell trajectories. Instead of the linear embedding tree, the new algorithm plans to apply a curved embedding tree with curved edges modeled by bounded principal curves. The details of the curved tree method can be found in Appendix B.

## 3. Simulation

### 3.1. Design

To check the accuracy and the robustness to the data noise of our algorithm, a *k*-nodes embedding tree in *p* dimensions is randomly generated with the embedding location of each node simulated from a multivariate normal distribution $N(0,I_{p})$. Considering the complexity in the real data set, additional noise in the data is generated from a known distribution. In the first part of the simulation study, the data noise is simulated from a standard normal distribution N(0,1). In the second part, the noise is generated from a student t distribution $t_{3}$. In this case, potential extreme outliers will occur, driving the data far away from the true tree structure. A parameter σ denotes the scale of noise in the simulated data and controls the noise level. The larger σ means more noise in the simulated data. In the study, we gradually increase the noise level σ to examine the robustness of our algorithm. Because of the significance of pseudotime in the real data application, the accuracy is calculated based on the ordering of pseduotime assignments. Kendall rank correlation coefficient is computed between the estimated pseudotime and the simulated pseudotime based on the true structure.

### 3.2. Comparison with Other Methods

Four existing single-cell trajectory-inferring methods are also applied to the simulated data as comparisons. These four approaches use an MST, the KNN graph or more complicated methods such as the principal curve in the trajectory-modeling part, which covers a majority of models for reconstructing a cell trajectory. Monocle models the trajectory by an MST in the lower dimension converted by ICA. More complicated than Monocle, Monocle2 uses an additional clustering algorithm to obtain the latent data representation in the lower dimension and then uses RGE to learn a principal curve. Slingshot also models the trajectory based on an MST. However, the MST is fitted on the clusters of cells and finally is treated as an initial guess for a simultaneous principal curve algorithm, which enables modeling of a nonlinear trajectory. SLICER first reduces dimensionality by LLE and then applies KNN graph for the trajectory-modeling step.

As shown in Figure 2, for the normal-distributed noise, all methods have the same trend that both the Kendall correlation and the stability will decrease as the noise level increases. Hence, the accuracy of the cell ordering based on the pseudotime is associated with the noise level. However, STS has the highest correlation and is the most stable one among all methods with any noise level. When the noise level is low, the correlation of our method is close to 1, so our method produces almost the same cell ordering result as the truth. SLICER has the second highest correlation, and Slingshot has poorer performance than SLICER. The two versions of Monocle methods have a similar performance with the lowest correlation among all methods. When the noise level is extremely high, all methods have a Kendall correlation lower than 0.5, and the estimated cell orderings are close to random assignments.

When the noise is generated from student t distribution in Figure 2, the Kendall correlations of all methods will drop faster than those in the normally distributed noise case when the noise level increases. The performance of all approaches is also more unstable than the normal case. With a low or moderate noise level, our method outperforms other approaches with the highest Kendall correlation. When the noise level reaches 1.5, the correlations of all methods drop below 0.5; thus, all approaches fail to accurately identify the true cell orderings. Monocle with DDRTree (Monocle2) is the most stable and least affected by the extreme outliers among all methods. STS has a slightly lower correlation than Monocle2 but is still compatible with other methods.

We also check the computational efficiency through simulations with different sample sizes and tree sizes. As Figure 3 shows, the computational cost increases nonlinearly with the growth in sample size when the tree size is fixed. However, there is a linear increasing trend indicated by Figure 3 for the running time when the tree size increases and the sample size remains the same. When the tree size is more than eight, the tree structure is complicated, making it difficult to estimate it through the data. In this case, STS will stop with a more straightforward estimated optimal tree with fewer edges.

### 3.3. Comparison between Linear and Curved Tree Methods

In order to check the accuracy and the robustness to data noise of our curved tree algorithm, a *k*-nodes curved tree with non-zero curvature parameters in *p* dimensions is randomly generated with the embedding location of each node simulated from a multivariate normal distribution $N(0,I_{p})$. Considering the complexity in the real data set, additional noise in the data is generated from a known distribution. Since the data is noisier in the curve tree framework, we only simulate the data noise from a standard normal distribution N(0,1). A parameter σ denotes the scale of noise in the simulated data and controls the noise level. The larger σ means more noise in the simulated data. In the study, we first gradually increase the noise level σ to examine the robustness of our algorithm when the curvature level is fixed. Then, we simulate the curvature parameters of the bounded principal curves from the normal distribution $N(0,cI_{p})$. The standard deviation *c* of this normal distribution controls the curvature of simulated curved trees. In detail, a larger *c* means that there is a higher curvature on average, and thus a more complicated tree is simulated. We also simulate data with different curvature levels when the noise level is fixed. We assess the performance of our method in two parts. To check the accuracy of cell orderings, the Kendall rank correlation coefficient is computed between the estimated pseudotime and the simulated pseudotime based on the true structure. In addition, we also check the accuracy of the estimated cell trajectory by the residual standard error. The residual standard error is calculated as the square root of the mean square error between the estimated data projections and the true projections. We also compare our curved tree algorithm with the linear tree algorithm.

As shown in Table 1, all approaches have the same nonlinear trend: both the Kendall correlation and the residual standard error with the associated standard deviation will decrease as the noise level increases. Hence, the accuracy of the cell ordering is based on the pseudotime and is related to the noise level, as we found for the linear tree method. Moreover, the accuracy of trajectory estimation is also related to the noise level. When we compare the two algorithms with the same noise level, there is no significant difference in Kendall correlation between the two methods. Regarding residual standard error, the curved tree algorithm outperforms the linear tree algorithm with much lower values. In general, the curved tree algorithm does not significantly improve the accuracy of cell orderings, but the estimation of cell trajectory is more accurate.

From Table 2, both the Kendall correlation and residual standard error are positively correlated with the curvature for the two methods. With the increasing curvature, there is no significant difference in Kendall correlations between the curved and linear tree methods. However, with the larger curvature, the Kendall correlations between the two methods have a more significant difference. In terms of residual standard error, the curved tree method has a smaller value than the linear tree approach with the same curvature. Moreover, with the increase in curvature, both two approaches become more unstable with the higher variation in Kendall correlation and residual standard error.

## 4. Application

### 4.1. Induction of Mouse Embryonic Stem (ES) Cell Differentiation

Time series data of 421 RamDA-seq samples with mouse ES cells were collected at five different time points with 157,717 gene features during the cell differentiation process. These time points are 0, 12, 24, 48 and 72 h after the induction of cell differentiation into primitive endoderm (PrE) cells [13]. We pre-processed the data in the same way as Cannoodt (2016) did. The data is first filtered by only keeping cells with good quality, so the number of cells reduces to 414. Then, the count data is normalized by log2 transformation. Finally, 23,658 gene features are selected based on the feature variability. With the final gene expression matrix, diffusion map is used to further reduce the data dimensionality. With the normalized data transformed into a lower dimension, we apply STS to estimate the cell trajectory and calculate pseudotime to order cells. Further, we also fit the same four existing TR methods in the simulation studies—Slingshot, SLICER, Monolce and Monocle2.

Hayashi (2018) identified a cell differentiation trajectory in their paper, where cells move from the initial state to the final state through an intermediate transition state. From Figure 4, our two approaches can obtain an estimated cell trajectory close to the truth. For the four existing TR methods, Slingshot, Monocle and Monocle2 cluster on cells but classify different numbers of clusters ranging from 5 to 16. Both SLICE and Monocle2 estimate the cell trajectory slightly differently from the other three methods, as they also identify a small branch at the transition state. In terms of pseudotime estimation, not all of these methods accurately order cells. Figure 5 shows the relationship between the estimated pseudotime and the true time for each method. For all the methods except SLICER, there is a positive correlation between the estimation and truth, which indicates the pseudotime calculation consistent with the true time points. SLICER is unable to estimate a pseudotime accurately related to the truth. The same result is indicated in Table 3. The Kendall correlation for SLICER is the lowest, at 0.39, among all TR methods. Other methods all have a Kendall correlation greater than 0.8 and our two methods have the highest correlation. All the methods except SLICER are able to estimate pseudotime accurately and recover the true cell orders.

Comparing our linear and curved tree methods, the curved tree method (0.0045) has a slightly lower residual standard error than the linear tree method (0.0055). Therefore, the curved tree method estimates the cell trajectory more accurately than the linear tree approach. In terms of pseudotime estimation, consistent with the findings from the previous data set, our curve tree method does not have any significant improvement on the cell orderings when compared with the linear tree method. Both the curved and the linear tree methods have the same Kendall correlation.

### 4.2. Resolution of Cell Fate Decisions from Zygote to Blastocyst

We also apply our STS methods on a single-cell expression data set with Ct values of 48 genes from 442 cells harvested over the first four days of mouse development. We try to recover the dynamic cell development process from zygote through blastocyst (from 1-cell stage to 64-cell stage). We follow the same data cleaning and normalization process as Guo et al. (2010) in [14]. Firstly, duplicate cells from two or more cell embryos are removed, and the cells with Ct values less than 28 are also removed. Then, we normalize the cell Ct values using the endogenous controls Actb and Gapdh by subtracting their average Ct values for each cell. With the final gene expression matrix, diffusion map is used to further reduce the data dimensionality. With the normalized data transformed into a lower dimension, we apply our linear and curved tree methods to estimate the cell trajectory and calculate pseudotime to order cells. Further, we also fit the same four existing TR methods in the simulation studies—Slingshot, SLICER, Monolce and Monocle2.

The previous study shows that the cells from the 64-cell stage are subdivided into the trophectoderm (TE), the epiblast (EPI) and the primitive endoderm (PE). In addition, the cells from the 32-cell stage can be classified as the inner cell mass (ICM) or TE in the middle of the cell development [14]. From Figure 6, our two approaches can obtain an estimated cell trajectory close to the truth. For the four existing TR methods, Monocle2 can also recover a similar three-branch structure to that in our method. However, Slingshot, SLICER and Monocle only identify two cell types at 64-cell stage. In terms of pseudotime estimation, not all of these methods accurately order cells. Figure 7 shows the relationship between the estimated pseudotime and the true time for each method. Both Slingshot and our approach show a positive correlation between the estimation and truth, indicating the pseudotime calculation consistent with the true time points. SLICER, Monocle and Monocle2 fail to distinguish between the first three stages as well as the last two stages. The same result is indicated in Table 4. The Kendall correlation for Monocle is the lowest, at 0.51, among all TR methods. SLICER and Monocle2 have a Kendall correlation equal to 0.69 and our algorithm has the second highest correlation of 0.76. Slingshot has a slightly higher correlation than our methods. This data set shows a more complicated dynamic process than the previous data set, so none of these methods can order the cells highly close to the truth. Nevertheless, STS and Slingshot can order the majority of the cells accurately.

Comparing between our linear and curved tree methods, the curved tree method (0.0220) has a slightly lower residual standard error than the linear tree method (0.0229). Therefore, the curved tree method estimates the cell trajectory more accurately than the linear tree approach. In terms of pseudotime estimation, consistent with the findings from the previous data set, our curved tree method does not have any significant improvement on the cell orderings when compared with the linear tree method. Both the curved and the linear tree methods have the same Kendall correlation.

## 5. Discussion

The reconstruction of a cell trajectory from the scRNA-seq data is considered as a nonlinear optimization problem for many existing TR methods. These TR methods are mainly based on an MST or KNN graph to search for an optimal solution locally. Instead of searching locally, our algorithm provides a novel approach to directly search a global optimal fitting tree. We apply a stochastic tree optimization algorithm after pre-processing the data with normalization and dimension reduction. Based on a penalized likelihood, we start searching with a one-edge-tree pool and then gradually move to a more sophisticated tree pool by adding a new node. Although we currently use squared Euclidean distance between the data point and its projection on the tree, the optimization function is flexible and can be extended to different forms. As the simulation study and two real data examples show, STS is more accurate in cell-ordering estimation and less sensitive to outliers and skewed expression distribution. Specifically, the global optimal tree search improves the estimation performance compared with local search methods. In general, STS can be applicable to reconstruct both a simple bifurcation trajectory and more complicated multifurcation trajectories, with both high accuracy in trajectory estimation and cell ordering.

As some other challenges introduced by the complex data structure remain, there are still several potential future extensions to our STS method. Firstly, the giant tree searching space encumbers the computational efficiency of the dynamic optimization process. Secondly, the current optimization step is sensitive to extreme outliers, though the extreme outliers do not frequently occur in the real dataset. To reduce the sensitivity to the outliers, the current sum square objective function could be replaced by other functions. Moreover, based on the likelihood function, we can also calculate the probability for each tree in the tree pool by assigning prior probability for the tree size. Then we can further extend to the inference part, such as hypothesis testing [15]. Finally, instead of the backward line search algorithm we can use other optimization algorithms to reduce the sensitivity to the initial tree guess. Nevertheless, all these extensions can enable our approach to be adapted to more complex and noisy real data sets.

## Figures and Tables

**Figure 1 genes-14-00318-f001:**
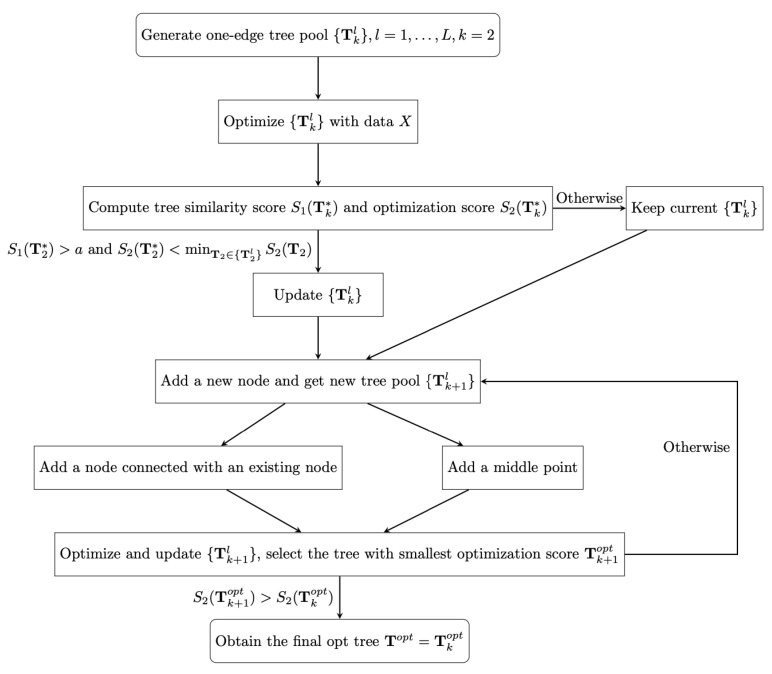
The flow chart for the stochastic tree searching algorithm.

**Figure 2 genes-14-00318-f002:**
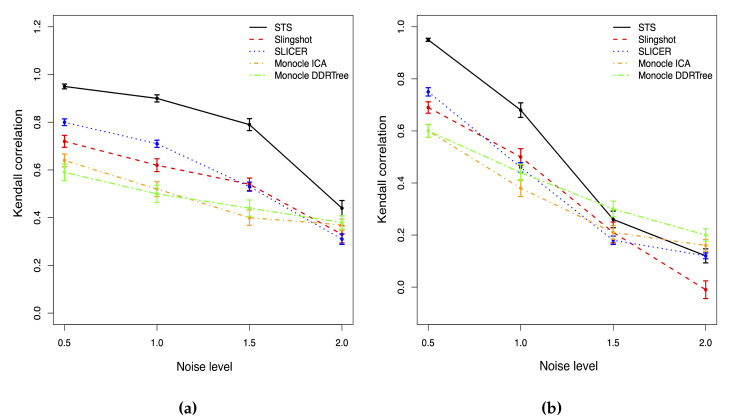
Kendall correlations with different noise level for two different noise distribution cases. (**a**) Normal. (**b**) T distribution.

**Figure 3 genes-14-00318-f003:**
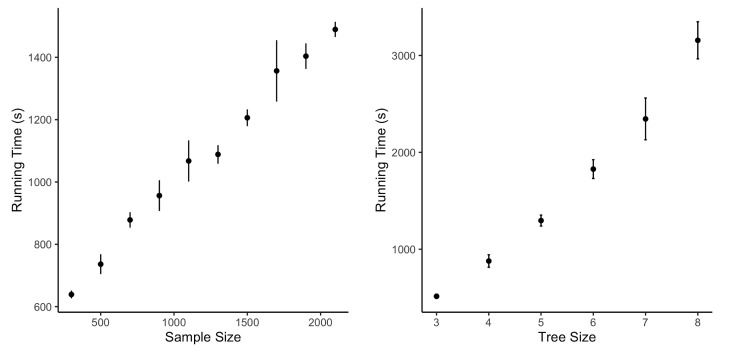
Running time with different sample sizes for tree sizes.

**Figure 4 genes-14-00318-f004:**
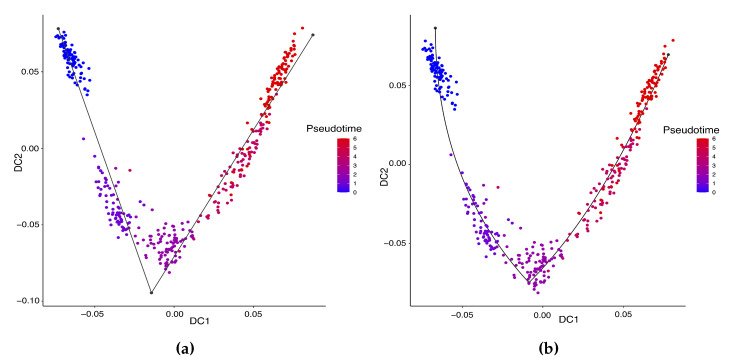
Estimated cell trajectories by our linear STS and curved tree methods on the mouse ES cell dataset. (**a**) Linear tree method. (**b**) Curved tree method.

**Figure 5 genes-14-00318-f005:**
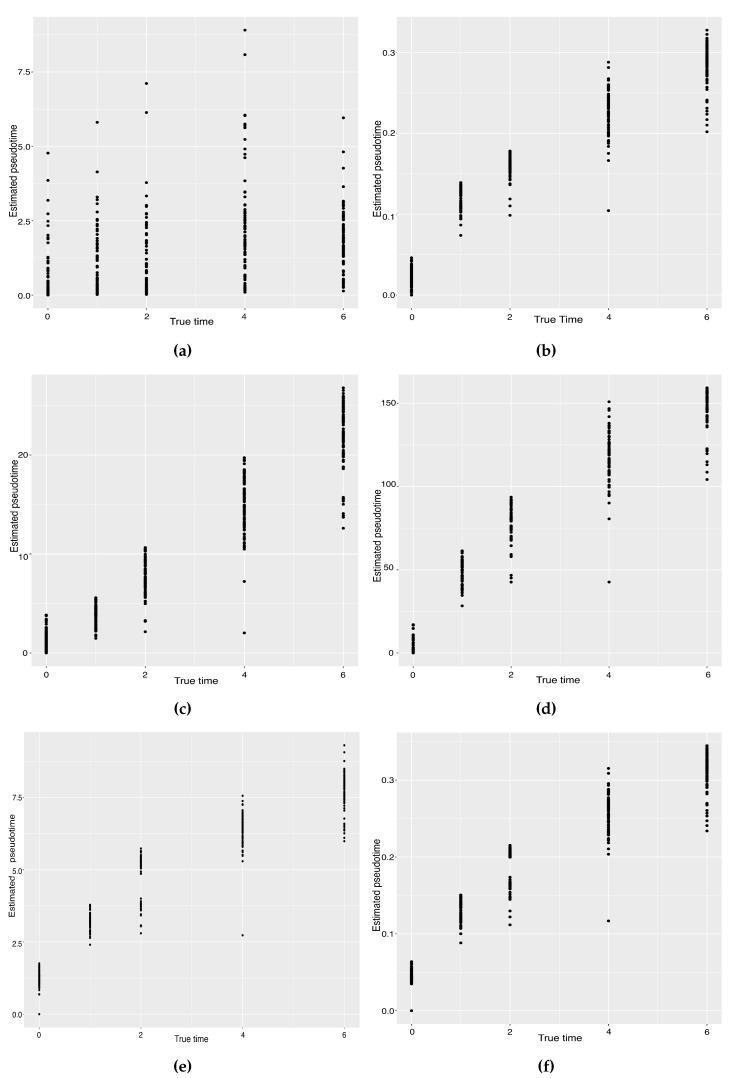
Scatter plots between the estimated pseudotimes and the true times for our approach and four existing TR methods on the mouse ES cell dataset. (**a**) SLICER. (**b**) Slingshot. (**c**) Monocle ICA. (**d**) Monocle DDRTree. (**e**) Linear Tree. (**f**) Curved Tree.

**Figure 6 genes-14-00318-f006:**
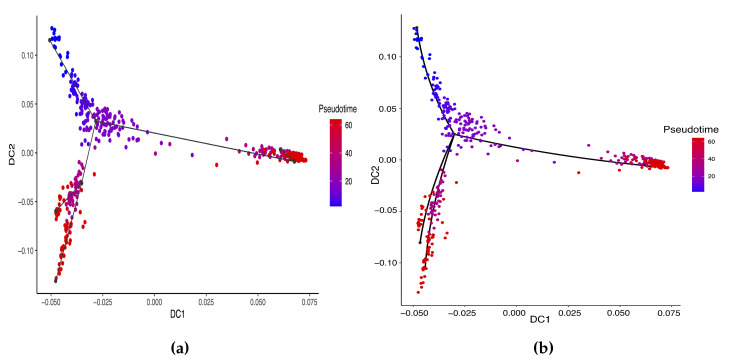
Cell trajectories estimated by our linear STS and curved tree methods on the Zygote–Blastocyst dataset. (**a**) Linear tree method. (**b**) Curved tree method.

**Figure 7 genes-14-00318-f007:**
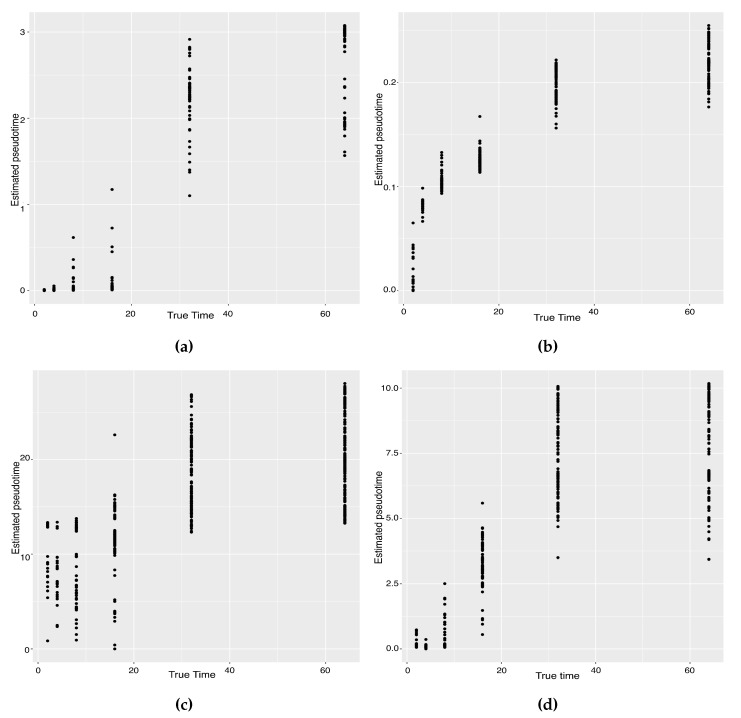
Scatter plots between estimated pseudotimes and true times for our approach and four existing TR methods on the Zygote–Blastocyst dataset. (**a**) SLICER. (**b**) Slingshot. (**c**) Monocle ICA. (**d**) Monocle DDRTree. (**e**) Linear Tree. (**f**) Curved Tree.

**Table 1 genes-14-00318-t001:** Mean Kendall correlations and mean residual standard error for both curved tree algorithm and linear tree algorithm with different noise level.

Noise Level	Mean Kendall Correlation (SD)		Mean Residual Standard Error (SD)	
	Curved Tree	Linear Tree	Curved Tree	Linear Tree
0.01	0.87 (0.23)	0.84 (0.26)	0.0356 (0.0325)	0.1812 (0.0365)
0.05	0.83 (0.29)	0.82 (0.29)	0.0529 (0.0293)	0.1839 (0.0399)
0.10	0.83 (0.25)	0.81 (0.29)	0.0889 (0.0285)	0.1988 (0.0410)
0.15	0.81 (0.27)	0.79 (0.30)	0.0889 (0.0283)	0.1978 (0.0403)

**Table 2 genes-14-00318-t002:** Mean Kendall correlations and mean residual standard error for both curved tree algorithm and linear tree algorithm with different curvature.

Curvature	Kendall Correlation		Residual Standard Error	
	Curved Tree	Linear Tree	Curved Tree	Linear Tree
0	0.85 (0.24)	0.84 (0.24)	0.0823 (0.0262)	0.1753 (0.0452)
0.5	0.81 (0.29)	0.81 (0.29)	0.0889 (0.0285)	0.1988 (0.0410)
1.0	0.83 (0.24)	0.78 (0.26)	0.1027 (0.0420)	0.2621 (0.0366)
1.5	0.60 (0.34)	0.57 (0.34)	0.1325 (0.0466)	0.3039 (0.0376)
2.0	0.48 (0.43)	0.43 (0.40)	0.1405 (0.0478)	0.3248 (0.0409)

**Table 3 genes-14-00318-t003:** Kendall correlation for all methods on the mouse ES cell dataset.

Method	Linear Tree	Cruved Tree	SLICER	Slingshot	Monocle ICA	Monocle DDRTree
Kendall Correlation	0.87	0.87	0.39	0.87	0.86	0.87

**Table 4 genes-14-00318-t004:** Kendall correlation for all methods on the Zygote–Blastocyst dataset.

Method	Linear Tree	Curved Tree	SLICER	Slingshot	Monocle ICA	Monocle DDRTree
Kendall Correlation	0.76	0.76	0.69	0.78	0.51	0.69

## Data Availability

The data and computer codes that support the findings in this paper are available at https://github.com/kkttzjy/STS. These data were derived from the following resources available in the public domain: https://www.ncbi.nlm.nih.gov/geo/query/acc.cgi?acc=GSE98664 accessed on 31 March 2016, and https://doi.org/10.1016/j.devcel.2010.02.012 accessed on 11 February 2022.

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
