# Peer review of "Reconstruction of Single-Cell Trajectories Using Stochastic Tree Search"

_genes, 2023, doi:10.3390/genes14020318_

Round 1

Reviewer 1 Report

I like the attempt of the authors and appreciate their effort to perform the research. However, the novelty of the work is not shown explicitly. I have accepted the manuscript for publication after some revisions.

 I have several comments with minor edits.

 1.       What is the novelty of the work? Please describe shortly.

2.       The objective of the study is not explained clearly in the manuscript. You should describe the objectives clearly.

3.       The axis ticks in several figures ( Figure 3-Figure 7) need to be shown clearly. Thus those figured need to be improved.

4.       Discussion and conclusion need to be elaborated.

 5.       I understood the authors discussed the results along with description of the findings.It would be nice if the authors concise the discussion with findings and explain more in the discussion section

Author Response

Please see attached response letter.

Reviewer 2 Report

Comments:

·      How does the authors construct the search trees? What are the variables for parent and child trees?

·      Did the authors compared their method with STREAM (Single-cell trajectories reconstruction, exploration and mapping of omics data with STREAM, Chen et al., nature communications 2019), update it with required information.

·      What are the method authors used for topology correctness? I haven’t seen information regarding topology characterization in the manuscript.

·      The nonlinear gene expression changes and the genes unrelated to the process lead to the possibility of branching trajectories, how did the authors address this challenging issue?

·      Are this method works for multivariate data? Please include relevant information.

·      Most of the figures require resolution and dimension adjustment, especially Figure 2, 4 and 6.

Author Response

Please see attached response letter.
